# Environmental data provide marginal benefit for predicting climate adaptation

**Forrest Li**[1,2]*, **Daniel J. Gates**[1,3], **Edward S. Buckler**[4,5], **Matthew B. Hufford**[6], **Garrett M. Janzen**[6], **Rubén Rellán-Álvarez**[7], **Fausto Rodríguez-Zapata**[7,8], **J. Alberto Romero Navarro**[4], **Ruairidh J. H. Sawers**[9], **Samantha J. Snodgrass**[1,6], **Kai Sonder**[10], **Martha C. Willcox**[10], **Sarah J. Hearne**[10], **Jeffrey Ross-Ibarra**[1,3,11]*, **Daniel E. Runcie**[2]*

1 Department of Evolution and Ecology, University of California Davis, Davis, California, United States of America, 2 Department of Plant Sciences, University of California Davis, Davis, California, United States of America, 3 Center for Population Biology, University of California Davis, Davis, California, United States of America, 4 Institute for Genomic Diversity, Cornell University, Ithaca, New York, United States of America, 5 United States Department of Agriculture-Agricultural Research Service, Robert W. Holley Center for Agriculture and Health, Ithaca, New York, United States of America, 6 Department of Ecology, Evolution, and Organismal Biology, Iowa State University, Ames, Iowa, United States of America, 7 Department of Molecular and Structural Biochemistry and Plant Sciences Initiative, North Carolina State University, Raleigh, North Carolina, United States of America, 8 Laboratorio Nacional de Genómica para la Biodiversidad/Unidad de Genómica Avanzada, Cinvestav, Irapuato, México, 9 Department of Plant Science, The Pennsylvania State University, State College, Pennsylvania, United States of America, 10 CIMMYT, El Batan, Texcoco, Estado de Mexico, Mexico, 11 Genome Center, University of California Davis, Davis, California, United States of America

* frrli@ucdavis.edu (FL); deruncie@ucdavis.edu (DER); rossibarra@ucdavis.edu (JR-I)

**Data availability statement:** The Github repo with analysis scripts is available at

## Abstract

Climate change poses a major challenge for both natural and cultivated species. Genomic tools are increasingly used in both conservation and breeding to identify adaptive loci that can be used to guide management in future climates. Here, we study the utility of climate and genomic data for identifying promising alleles using common gardens of a large, geographically diverse sample of traditional maize varieties to evaluate multiple approaches. First, we used genotype data to predict environmental characteristics of germplasm collections to identify varieties that may be pre-adapted to target environments. Second, we used environmental GWAS (envGWAS) to identify loci associated with historical divergence along climatic gradients. Finally, we compared the value of environmental data and envGWAS-prioritized loci to genomic data for prioritizing traditional varieties. We find that maize yield traits are best predicted by genome-wide relatedness and population structure, and that incorporating envGWAS-identified variants or environment-of-origin data provide little additional predictive information. While our results suggest that environmental data provide limited benefit in predicting fitness-related phenotypes, environmental GWAS is nonetheless a potentially powerful approach to identify individual novel loci associated with adaptation, especially when coupled with high density genotyping.

github.com/liforrest6/SeeD. Imputed GBS SNP data used for genetic analysis is available at https://data.cimmyt.org/dataset.xhtml?persistentId=hdl:11529/8702394. Intermediate results and data for plotting figures are available as a Dryad repository at https://datadryad.org/dataset/doi:10.5061/dryad.5hqbzkhhf. Curated information of field trial metadata and phenotypic BLUP data is available at https://data.cimmyt.org/dataset.xhtml?persistentId=hdl:11529/10548233 and benefits from this research are made available to shareholders and the public community as per the Standard Material Transfer Agreement (https://www.cimmyt.org/content/uploads/2019/04/SMTAFAQCIMMYTSept09.pdf).

**Funding:** FL would like to acknowledge funding from CIMMYT through the Gates Foundation (grant INV-030574) and the Jastro-Shields research award. Additionally, the authors would like to acknowledge funding from the NSF: award number 1546719 to JRI and award number 1238014 to ESB. JRI acknowledges support from USDA Hatch project CA-D-PLS-2066-H, RRA acknowledges support from USDA Hatch Project 02776, and DER acknowledges support from USDA Hatch Project 1010469. The funders had no role in study design, data collection and analysis, decision to publish, or preparation of the manuscript.

**Competing interests:** The authors have declared that no competing interests exist.

## Author summary

Populations of natural and cultivated plant and animal populations will be affected by more extreme climate events such as drought and flooding in the future. We explore whether characterization of the environment-of-origin of each accession in a large sample of traditional maize germplasm can be used to accelerate conservation and breeding efforts for adaptation. We compare the utility of genotype and environmental data for predicting fitness of individuals in a number of common garden trials. We find that environment-of-origin data and genome scans for loci that associate with abiotic environmental variables provide surprisingly little benefit to prioritizing accessions for improvement, despite clear evidence of environmental adaptation in these accessions. These results provide important practical insight into the use of gene banks for climate adaptation.

## Introduction

Protecting plant and animal populations against the harmful effects of climate change is one of the most important goals of genetic research today. Global temperatures have increased 1.5°C above pre-industrial levels [1], with further increases of 3.3-5.7°C possible in the next 100 years [2]. Models under future warming scenarios indicate yield losses across major staple commodities in the range of 25-50% by 2100 for crops [3,4], as well as for ruminant livestock and aquaculture [5–7]. Likewise, natural populations of plants have already been affected by rising temperatures [8,9]. Restoration and conservation of natural plants and animals will require identifying individuals that can persist in future environments.

One promising strategy for improving the resilience of populations to climate change is to harness existing genetic variation to identify sources of abiotic stress resistance [10–13]. Locally adapted natural populations of many plant and animal species contain substantial genetic diversity [14–22] that can be linked to environmental adaptation and utilized for conservation programs [13]. Similarly, traditional varieties (*i.e.,* landraces) of many cultivated plant and farmed animal species have been evolving in diverse environments for thousands of years [23–27], resulting in a broad base of genetic variation and adaptation to a wide range of environmental niches.

Incorporating beneficial diversity into existing populations is a difficult endeavor. This has most commonly been done in plant breeding, where breeders have first identified traditional varieties with apparently useful characteristics, then backcrossed these against elite germplasm to introgress adaptive alleles while maintaining the agronomic performance afforded by the genetics of elite material in the rest of the genome [28–30]. Because this is difficult to do comprehensively with tens of thousands of traditional varieties, efforts to prioritize traditional varieties based on their environment-of-origin have gained traction [31,32]. Here, a sample of traditional varieties is first grown in common garden field trials to identify beneficial traits, then trait presence is modeled as a function of environmental variables to identify other traditional varieties in the full collections that may possess similar traits. This approach has been successfully applied to identify drought adaptation in broad bean, flower frost tolerance in wild strawberry, and seed traits in bread wheat [32–34]. A major limitation of this approach, however, is that it does not consider genetic information. If beneficial exotic alleles are not specifically tracked and prioritized using genetic markers, they are rapidly lost when backcrossed into elite material [35,36].

Molecular markers can be used to track and incorporate individual loci of interest. Early use of molecular markers in breeding used QTL mapping and marker assisted selection to select candidate varieties and introgress specific alleles [37,38]. With the increased accessibility of whole genome marker data, genomic prediction approaches have become widely used in breeding to make selections among candidate varieties [28,39]. As genome-scale data become available for large numbers of individuals from natural populations or traditional crop varieties in germplasm repositories, genomic prediction of phenotypes relevant for climate adaptation is now possible both for breeding [35,40,41] and conservation approaches such as assisted gene flow [42]. But implementing genomic prediction is challenging as it requires collecting sufficient high quality training data on important traits in relevant environments.

Environment-of-origin data and genome-wide marker data are complementary, however, and a prediction approach that incorporates both data could provide additional advantages [34,43]. One approach to do so involves using geographic occurrence to predict optimal genotypes for future climates [44], but these predictions often correspond poorly to experimental [45] and simulated [46] evaluations of phenotype. Alternatively, environmental and genomic marker data can be combined to identify specific loci associated with adaptation. If populations are adapted along environmental gradients, then the frequencies of loci underlying adaptation may show correlations with those gradients. Genome-wide scans for alleles with frequencies correlated to environmental gradients are called environmental genome-wide association studies (envGWAS) [47], and have been applied in natural populations such as Loblolly pine and *Medicago* as well as traditional varieties of crop species such as maize, barley, and soybean [48–52]. Such scans may be important for predicting phenotypes in future environments. In *Sorghum*, for example, [43] showed that alleles associated with drought and low pH successfully predicted variation in plasticity in field trials.

Maize (*Zea mays spp. mays*) is the most productive crop globally and has adapted to grow in nearly every environment that humans inhabit [53], yet yield in modern maize lines has already been impacted by climate change [54]. Traditional maize varieties and wild teosintes show clear patterns of adaptation along abiotic gradients [18,23,55–57] and constitute an extensive but largely untapped well of genetic diversity for plant breeding [58,59]. For example, traditional maize varieties grown in common gardens across multiple elevations in Mexico demonstrated substantially higher fitness when grown in trials more similar to their native environments [60], and flowering time, macrohair, and anthocyanin traits showed evidence of adaptive divergence in reciprocal transplant experiments across highland/lowland environments using a broader population of North and South American accessions [61]. Adaptation in maize and its wild relatives has resulted in selection on a number of individual loci [18,57], many of which show strong associations between genotype and the abiotic environment, including inversions associated with flowering time [50,62] and macrohair growth [63], as well as individual genes that regulate pospholid metabolism and flowering time [64]. While adaptation along an environmental gradient does not necessarily translate into local adaptation—where an individual has highest relative fitness in its native environment—adaptation along abiotic gradients is likely more relevant for predicting evolutionary changes associated with changing climates. Maize may therefore be an ideal candidate for a strategy that combines modern detection methods of adaptive variation with novel technologies to minimize the challenges of using exotic germplasm. CIMMYT curates the world's largest collection of over 24,000 traditional maize varieties, a resource that could play a crucial breeding role in mitigating the negative effects of climate change on maize. With such a large and diverse collection, narrowing down favorable material adaptive to target climates to introduce into pre-breeding programs is a difficult task.

In this study, we identify adaptive genetic variation in a core set of CIMMYT's maize germplasm collection. We evaluate methods for jointly using genomic and environmental data to estimate the utility of environmentally associated alleles in predicting the genetic value of traditional varieties. First, we show that traditional maize varieties are adapted along multiple environmental gradients using large-scale field trials and by measuring the genome-wide correlation between genetic variation and environmental gradients. Next, we perform envGWAS to identify individual genetic markers associated with environmental gradients and find dozens of potentially adaptive loci. Finally, we evaluate the utility of envGWAS-identified markers and environmental data for prioritizing useful traditional varieties compared to conventional genomic prediction methods.

## Results

### Phenotypic trials demonstrate environmental adaptation in maize

To be able to test different approaches for identifying genetic diversity useful for agronomic improvement, we used a panel of $\approx 4,100$ traditional maize varieties from CIMMYT's maize seed bank (Fig 1). Of these, GBS genotyping [65] and growing-season environmental data from their collection location were available for 3511 genotypes representing 2895 germplasm bank accessions, which were used for genotype-environment associations. We combined these data with phenotypes from the CIMMYT Seeds of Discovery (SeeD), consisting of a set of 23 common garden trials representing 13 locations over multiple years at varying elevations [see S1 Table][41,50,62]. We evaluated phenotypic data from crosses with common testers in multiple common gardens across Mexico, totaling more than 280,000 progeny and spanning almost 20,000 plots (see **Materials and Methods**). We focused on four yield components (see S3 Table): grain weight per hectare (GWPH), field weight (FW), plant height (PH), and bare cob weight (BCW), measured on 144 to 1434 (mean = 586) testcross families per trial (S2 Table). In total, testcross families representing 2518 accessions that had genotype, climate, and phenotype data were used for phenotypic analysis.

To test for environmental adaptation in our sample of traditional varieties, we modeled the correlations between yield traits—a good proxy for fitness in domesticated plants—and the annual environmental values of each accession's collection location (S5 Table). We predicted that if traditional varieties are adapted along an environmental gradient, then for each common garden, traditional varieties collected from extreme values of that environmental factor should show reduced yield. We further predicted that the environmental value of varieties with the highest yield in each trial should positively correlate with the environment of the common garden in which they were grown. We fit quadratic curves to yield-related trait values along three environmental gradients (elevation, mean annual temperature, and total annual precipitation) and estimated the environment-of-origin value corresponding to the apex of these curves for each trial, henceforth the optimum environmental value. We then regressed these optimum environmental values against the real environmental values across all trials using a Bayesian model, and estimated the slope from the posterior distribution of these regressions.

For all yield traits, the posterior mean of the slope parameter was positive for elevation, precipitation, and temperature, indicating that adaptation occurs along these environmental clines (S6 Table). 95% posterior intervals of the slope parameter included the value $\beta_h = 1$ (equivalent to a perfect one-to-one relationship between trial environment and estimated optimal environment) for field weight along elevation ([0.34,1.22]), precipitation ([0.17,4.07]), and temperature gradients ([0.02,1.13]). This indicates that for field weight, our results are consistent with an increasing and correlated relationship between real and estimated

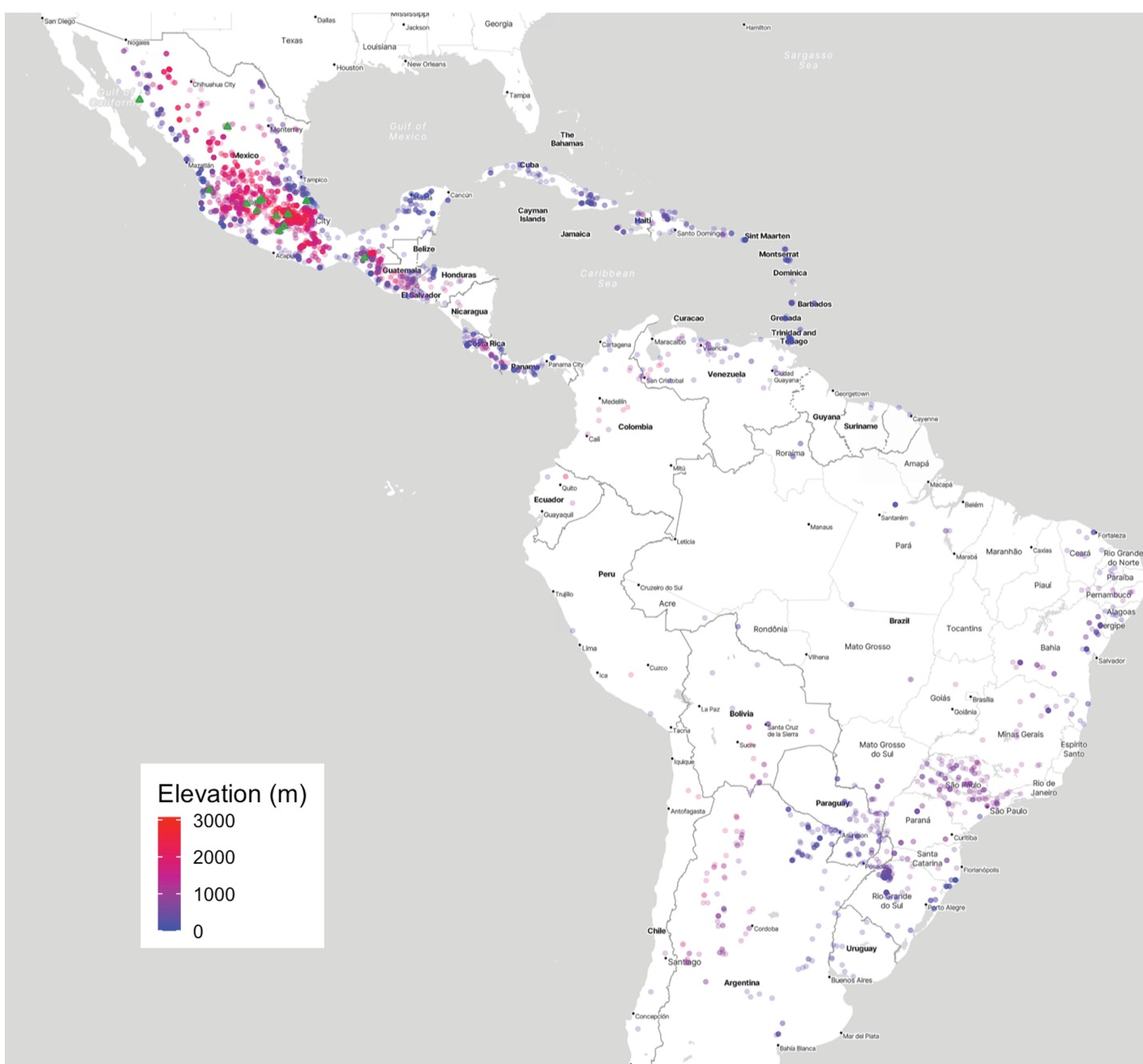

**Fig 1. Map showing the 2895 unique geo-referenced locations of traditional maize variety accessions from CIMMYT used for envGWAS, colored by elevation.**
Green triangles in Mexico indicate locations where field trials were held. Base layer map tiles are from Stamen Design under CC BY 4.0, accessed via the R package ggmap [66]. Data by OpenStreetMap, under ODbL.

optimal environment, providing strong evidence of environmental adaptation along all three clines (Fig 2). For grain weight per hectare, posterior means of the slope were smaller than one, but still positive, providing moderate evidence for adaptation along these environmental clines. However, posterior interval widths for this trait were much larger, because this trait was measured in fewer trials.

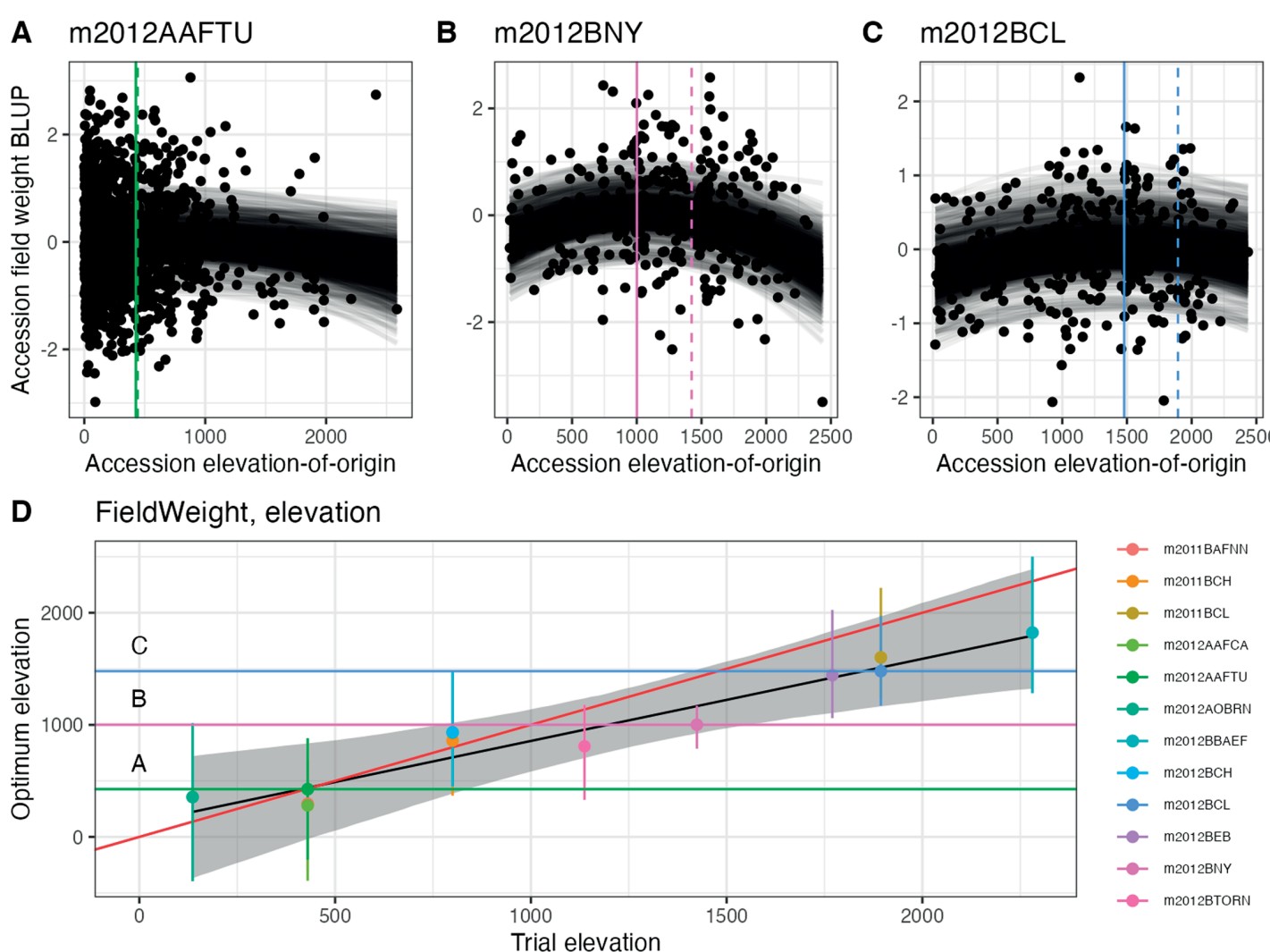

**Fig 2. Phenotypic trials show evidence of environmental adaptation to elevation in field weight. A–C)** Elevation transfer distance plots for three selected trials: **(A)** m2012AAFTU, **(B)** m2012BNY, **(C)** m2012BCL in Agua Fria, San Pedro, and Celaya in 2012, respectively. Points in each panel reflect scaled BLUPs of phenotypic values of field weight for each maize accession grown in the trial as a function of the elevation-of-origin of the accession. Curves reflect posterior draws of the model fit in each trial. Solid vertical lines represent the posterior mean estimate of the optimal elevation-of-origin for accessions grown in that trial, while dashed vertical lines represent the real elevation of the trial. **(D)** Posterior density of the regression of optimal elevation-of-origin against trial elevation. Points and vertical lines represent posterior means and 95% posterior intervals around the optimal elevation-of-origin for each trial. To aid comparison to the trials in panels (A–C), horizontal lines corresponding to the optima in each trial are drawn. The black line and grey ribbon show posterior mean and 95% posterior credible intervals of the relationship between optimal elevation and the trial's elevation. The red line shows a 1-1 linear relationship for comparison.

For bare cob weight and plant height, while we observe positive posterior means for slope for all environmental variables (S6 Table), 95% posterior intervals were not consistent with a slope of one, save for plant height along precipitation ([0.20587697,1.9036396]), indicating that while there is a correlation between the optimal environment and the true environment for these traits, traditional varieties may be under-adapted for these traits to the range of environments we tested here. Interestingly, we see that the posterior mean slope for plant height on precipitation is close to one (S6k Fig), but the intercept was much higher than zero. This

suggests that the optimal precipitation for a trial for maximizing plant height is always much higher than the actual trial's precipitation value.

## Genotypic variation across environment provides further evidence of local adaptation

If traditional varieties have adapted along environmental gradients as suggested by our trial data, we should likewise observe broad scale correlations between genotype and environment. To test this, we created a dataset of growing season environmental variables for 2895 unique collection locations (see **Materials and Methods**). We then matched these environmental data to samples with available genotyping by sequencing (GBS) data for a total of 3511 genotypes [50] at 345,270 imputed biallelic SNPs with minor allele frequency > 0.01.

To directly estimate how well genotype data alone predicts environmental variation among accessions, we built a genomic prediction of environment (GPoE) model for all environment variables using MegaLMM [67] and evaluated its accuracy using 10-fold cross-validation (Fig 3). GPoE predictive ability was high for elevation (average Pearson's $r$ = 0.839), temperature range ($r$ = 0.838), and minimum temperature ($r$ = 0.829), and moderate for other variables (0.637 < $r$ < 0.761; see Fig 3).

While environmental adaptation predicts a strong correlation between genotype and environment, an alternative explanation is that environmental variation is spatially autocorrelated at a scale greater than typical dispersal. Under such a scenario, even in the absence of adaptation, individuals would be found in environments highly similar to their relatives, and

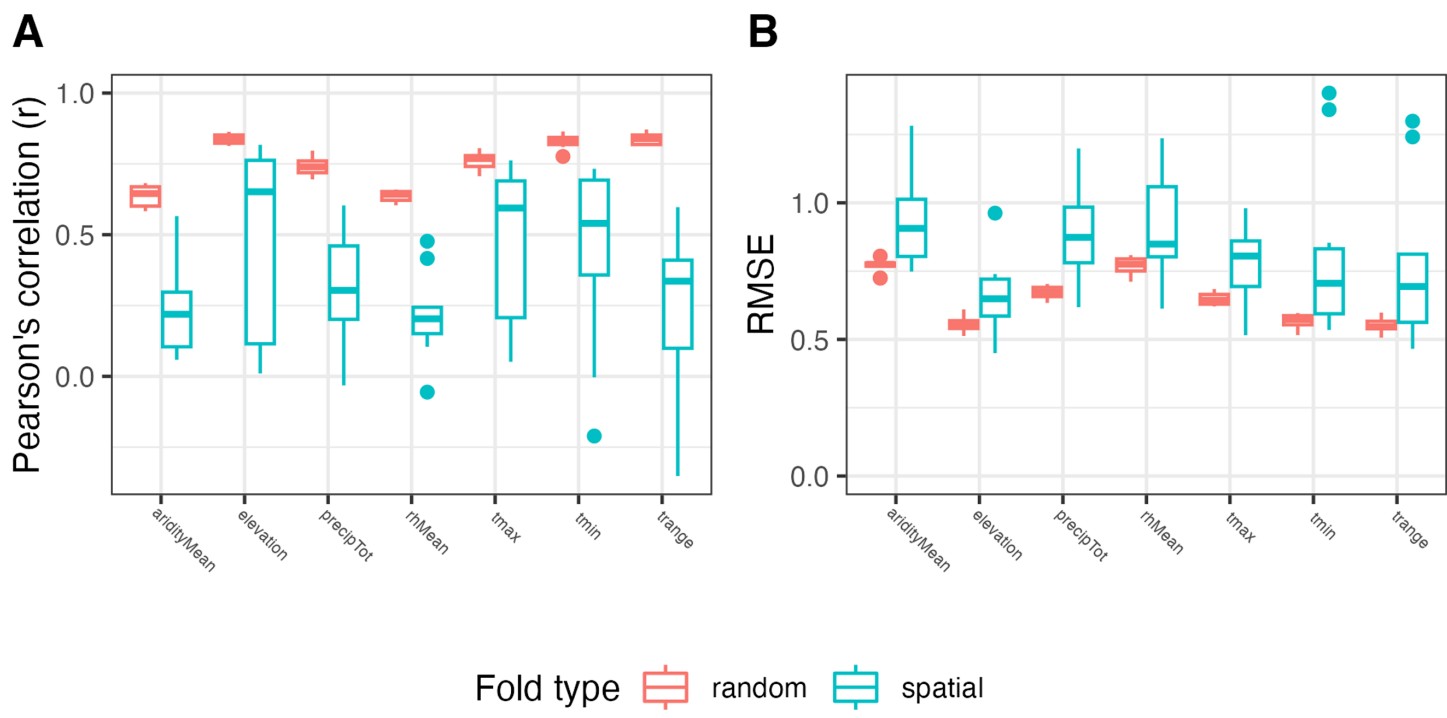

**Fig 3. Correlations between genetics and environment are demonstrated through prediction but decrease when considering spatial distance.** For each environmental variable, the **(A)** Pearson's $r$ correlation and **(B)** root mean square error (RMSE) between observed and predicted environment values from the GPoE model of random (salmon) and spatial (cyan) cross-validation folds.

thus environments of collection would be predictable from genetic relationships. To distinguish these possibilities, we split our study population into 10 spatially separated sub-samples (see **Materials and Methods**) and repeated the above GPoE analysis, predicting environmental values of individuals from one geographic region using only individuals from other geographic regions to train our models.

Predictive ability for spatially distant accessions is still greater than zero for all environmental variables ($P < 0.001$) but is significantly lower ($P < 0.001$) and has higher RMSE ($P < 0.001$) and higher variation among folds than for randomly sampled accessions (Fig 3).

## Gene-environment association identifies candidate loci

Despite the lower predictive ability of spatially explicit models, our GPoE analysis points to a meaningful association between genetic variation and environmental variation. To identify loci contributing to this relationship, we employed a multitrait environmental GWAS (envGWAS) implemented in JointGWAS [68] to find SNP markers associated environmental variation. Our envGWAS model controls for residual correlation between environmental factors and population structure and prioritizes SNPs associated with at least one of the environmental variables (see **Materials and Methods**).

We found 330 SNPs with a p-value less than $10^{-5}$ (Fig 4A), which we grouped into 32 independent loci based on LD. For most of these candidate loci, linkage disequilibrium decays within several Kb, with notable exceptions of *Inv4m*, a large 13Mb structural variant located

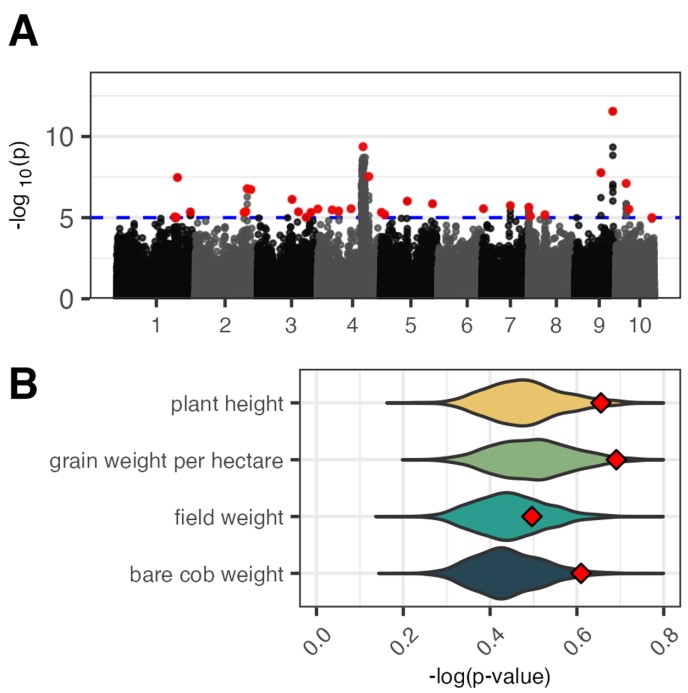

**Fig 4. Multivariate envGWAS and phenotypic enrichment. A)** Manhattan plot of multivariate envGWAS results testing each SNP for association against the set of seven climatic variables. Top SNPs representative of significantly associated loci in a nearby LD block are highlighted in red. Cutoff threshold of $10^{-5}$ represented by blue dashed line. **B)** Comparison of the average strength of yield trait phenotypic associations for the set of 32 top SNP hits selected by the envGWAS (red diamonds) against 1000 sets of 32 random SNPs of matching allele frequency (violin density plot). X-axis represents mean -log(p-value) of association to the given phenotypic trait for all SNPs in each set.

on chromosome 4; the *hsftf9* locus at SNP S9_148365695, with significantly associated SNPs 250kB away in LD; and SNPs at S9_101922947 and S10_41085126 (S13 Fig).

Although we find fewer loci than previous envGWAS of these data [50,62], our approach likely has a lower false positive rate (compare S10 Fig to Fig S14 in Romero-Navarro et al. [50]) due to our controlling for genetic relatedness among traditional varieties, and the way we accounted for multiple testing issues when running envGWAS on multiple correlated environmental variables. For example, we identify loci including *Inv4m* and a candidate SNP on chromosome 9 near an annotated heat-shock-related transcription factor *ZmHs-ftf9* (Zm00001d048041) [62]. The *Arabidopsis* ortholog of this transcription factor (AtHSF1) is expressed under heat stress and is responsible for activation of heat-shock-protein-based thermotolerance [69,70], and the tomato homolog HSFA1 acts as a master regulator of thermotolerance [71]. In maize, *ZmHsftf9* has been shown to have differential transcript splicing between high and low temperature regimes [72] and is associated with yield differences under drought in African varieties [62].

## Are genotype-environment associations useful for breeding?

Given these phenotypic and genetic signals of environmental adaptation, we asked whether the loci that we discovered could help breeders select useful traditional varieties.

As a first test of the value of envGWAS loci, we asked whether as a group these loci were significantly associated with phenotypic variation in our trials. Compared to random SNPs matched for minor allele frequency, envGWAS SNPs showed stronger associations to bare cob weight ($P = 0.023$), grain weight per hectare ($P = 0.02$), plant height ($P = 0.037$), and days to flowering ($P = 0.02$), but not for anthesis silking interval ($P = 0.969$) or field weight ($P = 0.275$) (Fig 4B).

To evaluate if these associated loci could explain phenotypic variation directly in genomic selection contexts, we estimated the percentage of variance in the yield trait BLUPs that could be predicted by these 32 loci (Fig 5). Since these loci may indirectly provide information on genome-wide population structure, we compared models with the top 32 envGWAS SNPs augmented by the first five principal components (PC) of the genomic relationship matrix (GRM) to a base model only using those PCs, as well as an alternative model including the PCs and random sets of SNPs matched by allele frequency with the envGWAS SNPs. The first two PCs were associated with elevation (Pearson's correlation $r = 0.83$) and latitude of origin ($r = -0.58$) respectively (S6 Table), but together explained less than 5% of the genotypic variance (S2 and S3 Figs). We assessed predictive ability only within sets of testcross families sharing the same hybrid tester so that predictive abilities would not be inflated by differences in genetic values among testers. The model including just the first five principal components had the highest predictive ability across trials and tester for field weight ($r = 0.305$), followed by grain weight ($r = 0.268$), plant height ($r = 0.207$), and bare cob weight ($r = 0.155$) (Figs 5 and S8). We found little evidence that including envGWAS SNPs in our models improved prediction accuracy for any of the yield traits beyond that captured by the genetic PCs ($P = 0.999$ for field weight, $P = 1.00$ for bare cob weight, $P = 0.999$ for grain weight, and $P = 0.997$ for plant height), and little evidence that they improved prediction over randomly selected SNPs ($P = 0.999$ for field weight, $P = 0.999$ for bare cob weight, $P = 0.9963$ for grain weight, and $P = 1.00$ for plant height).

We compared these models to two alternative strategies for predicting the value of traditional variety alleles for increasing yield. We fit genomic prediction models using all ~300K GBS SNPs as well as environment-of-origin prediction models using the data associated with each traditional variety. The first model evaluated the total value of all (common) alleles from

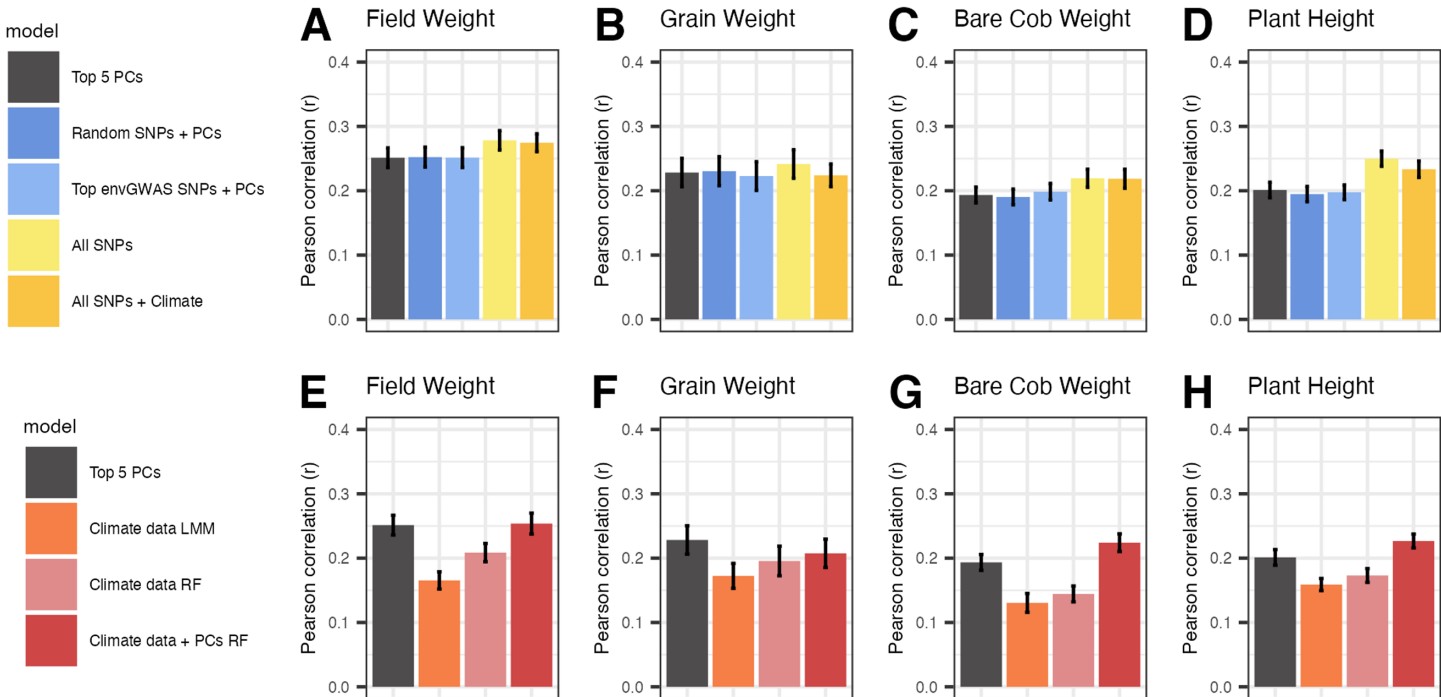

**Fig 5. Genomic and environmental prediction models.** Comparison of prediction models predicting field weight, grain weight, bare cob weight, and plant height. Y-axis represents predictive ability (Pearson's *r*) between a traditional variety's predicted value and actual BLUP value. Bar plots represent mean predictive ability across all tester families, and averaged across all trials, colored by selected genomic models (**A–D**) and environmental models (**B–H**).

the GBS data irrespective of their environmental distribution. The second model evaluates the gain in predictive ability provided by environmental data beyond that already captured by genetic information.

The genomic prediction model using all SNPs explained significantly more phenotypic variation in field weight than the PC-only ($P = 0.045$) and envGWAS SNPs + PC models ($P = 0.013$). We also found the all-SNPs model to have higher predictive ability for bare cob weight ($P = 0.001$ and $P = 0.001$ relative to PC-only and envGWAS SNPs + PC, respectively) and plant height ($P < 0.001$ and $P < 0.001$, respectively), but the improvement was not significant for corrected grain weight ($P = 0.228$ against the PC-only model and $P = 0.098$ against the envG-WAS SNPs model) In contrast, environment-of-origin prediction models that included PC features did not significantly increase predictive ability over the PC-only model ($P = 0.999$ for field weight, $P = 0.997$ for bare cob weight, $P = 1.000$ for grain weight, $P = 0.998$ for plant height), and the environment-of-origin prediction models that included all SNPs did not significantly increase predictive ability over the genomic prediction model ($P = 0.999$ for field weight, $P = 0.997$ for bare cob weight, $P = 1.000$ for grain weight, $P = 0.998$ for plant height).

To assess whether correction for population structure may have weakened our ability to identify adaptive alleles, we removed population structure correction from our envGWAS. We obtained 1064 lead SNPs representing 5014 SNPs associated ($P < 10^{-250}$) to climate, and used these to derive new SNP-based prediction models for each phenotype. When supplemented with genomic PCs, models based on these newly selected SNPs occasionally performed better than PCs alone, and sometimes better than random matching SNPs, but still did not perform better than the gBLUP model using all SNPs (S11 Fig).

These results consistently show that environment-of-origin data provide little additional benefit for predicting variation in yield traits among traditional varieties beyond that captured by GBS markers. However, environment-of-origin data is still more accessible than GBS data for many germplasm collections. To test whether environments-of-origin are useful in this dataset when GBS data are not available, we assessed the accuracy of predictive models using only environment-of-origin data in linear models and random forests (Figs 5B and S9). Predictive abilities of these environment-of-origin only models were significantly greater than zero on average ($P < 0.001$ for all traits), but generally had low accuracy, regardless of the methodology chosen.

## Discussion

A major goal of both breeding and conservation programs is to identify individuals and genetic variants to improve populations to be more resilient to the effects of climate change. We hypothesized that existing germplasm collections of traditional varieties, collected across a wide range of climate gradients, carry useful diversity for adapting existing varieties to new environments. Using a large panel of geo-referenced and genotyped maize collections, we evaluated whether environmental data could be used to prioritize traditional varieties for improving fitness traits in breeding. Despite the fact that these accessions exhibit both genomic and phenotypic evidence of environmental adaptation, we found that incorporating environmental data from their collection locations provided little additional information directly in genomic breeding.

On the one hand, our results are encouraging in that we find convincing evidence of environmental adaptation in our field trials. To address a confounding effect common to common garden study designs, where only phenotypes of surviving individuals are available and unfit offspring are not measured [73], traditional varieties were crossed mostly to hybrid testers adapted to similar elevations and were grown at these elevations. While allowing us to assess a greater number of traditional varieties, this choice likely attenuated some signal of environmental adaptation in our experiment. Despite this, our results show that environments-of-origin of traditional varieties are correlated to yield traits (Fig 2) and thus should be informative when considering agronomic performance across diverse environments. Likewise, even after accounting for spatial autocorrelation and population structure, our GPoE analyses find meaningful correlations between predicted and observed environments, pointing to a genotypic signal of local adaptation. Consistent with previous analyses of these data [62], we identify a number of convincing outlier loci, including *Inv4m*, a large inversion introgressed from *Zea mays spp. mexicana* [18,74] that appears to impact flowering time [50], and the heat shock protein *hsftf9* that shows differential splicing under different heat regimes [72]. Combined, the envGWAS-discovered loci are enriched for associations with phenotypes of agronomic importance (Fig 4) indicating that envGWAS successfully identifies loci of functional relevance.

Despite this, while we can identify environment-associated alleles from envGWAS that are enriched for phenotypic relevance, our results suggest that the specific loci that we discovered by envGWAS do not directly improve predictions of phenotypic values of traditional maize testcrosses in modern agronomic contexts and environmental characterization of traditional maize origins appears only weakly predictive of genetic values for breeding.

### The potential and limitations of envGWAS

envGWAS is an appealing technique for discovering candidate loci driving local adaptation to environment gradients because it can be applied to large collections of traditional germplasm

without the need for expensive common gardens or field trials. However, like any GWAS method, envGWAS cannot be comprehensive – it will likely miss many of the most valuable alleles (false negatives) and can also be subject to high false positive rates.

There are several possible reasons why envGWAS may miss important loci associated with environmental adaptation. First, GWAS prioritizes variants based on the amount of variance in the trait of interest, where the amount of variance explained by a given locus is proportional to the product of its effect size and allele frequency. Models of adaptation to changing climates suggest polygenic architectures consisting of many loci of small effect may be more successful than oligogenic architectures governed by large-effect loci [75,76]. Simple models of stabilizing selection [77] or polygenic adaptation to a new optimum [78] also predict that, when present, large effect alleles will be kept at low frequencies. While some models of adaptation with gene flow show that selection can favor high frequency, large effect alleles [79], these predictions break down for sufficiently polygenic traits [80]. The available evidence suggests that phenotypes [81] and climate adaptation [50,62] in maize are highly polygenic, suggesting that association approaches may miss loci of small effect or large effect loci that are rare in the population.

Second, because LD decays quickly in populations of traditional maize varieties [82], the ability of GWAS approaches to identify causal genes or QTL from sparse sequencing like GBS is impacted [83]. The $\approx 350,000$ GBS markers in our dataset are not distributed uniformly across the genome and constitute only a small fraction of the common variants in maize [84]. Higher-density genotyping of this population would likely help discover more causal loci. Finally, even if the loci driving adaptation in these diverse panels had sufficiently large effects, sufficiently high allele frequency, and were in LD with GBS markers such as to be detectable by envGWAS, adaptation may be controlled by latent environmental variables that we did not measure or are otherwise unknown. We included seven environmental variables describing variation in climate relevant to plant growth and development in specific growing seasons. However, there are many other environmental and management variables that affect plant performance that we did not include, such as soil variables, planting densities, and biotic pressures [85]. Alleles that increase performance along these gradients may have contributed to environmental adaptation or be useful in certain agronomic contexts but would have been missed by our analysis.

At the same time, envGWAS may also be sensitive to mechanisms that create false-positive associations that do not affect fitness for agronomic or conservation applications. While false positives may be common due to poorly calibrated statistical tests that run envGWAS independently for each variable, we use MegaLMM and JointGWAS [67,68] to de-correlate genotype-environment and environment-environment relationships, resulting in well-calibrated statistical tests which should reduce our false positive rate. Loci that improved fitness in historical contexts (i.e. traditional smallholder or non-irrigated agriculture), but do not control for fitness in trial environments can also be considered false positives because they are not useful for future adaptive potential (i.e. increases in yield in the breeding context).

We might find such loci if they controlled responses to environmental variables that were correlated with any of our seven focal environmental variables across traditional farms. For instance, since biotic stresses such as disease are correlated with rainfall or humidity [86], loci underlying disease resistance may be discovered as having an association with precipitation level, despite the causal relationship being disease pressure on the locus. Finally, population structure and spatial autocorrelation of environmental variables can generate non-adaptive associations between neutral alleles and environmental variables and are known issues in landscape and conservation genomics [47,87,88].

In addition, population structure exists within maize traditional varieties for many reasons including dispersal patterns shaped by geographic distance, human migration, farmers' preference for quality traits, seed sharing via trade, and introgression of wild relative alleles in limited geographic regions [89–92]. If populations also occupy different environmental zones, those divergent alleles will also associate with environmental gradients. We attempted to reduce the number of such false positive discoveries using an estimated genomic relationship matrix as a covariate in our envGWAS, which greatly reduced the number of significant discoveries relative to earlier analyses of this dataset [50,62]. However, this correction will also restrict the discovery of any alleles that drive population differentiation, including true positive adaptation to those environmental gradients. To mitigate this effect, we used a leave-one-chromosome-out (LOCO) strategy in calculating the GRM, which strikes a balance between sensitivity (reducing false negatives) and specificity (reducing false positives) due to population structure [93].

Future efforts that model genetic background and migration by accounting for spatial effects [94] may improve the true discovery rate of envGWAS.

## Evaluation of alternative strategies for discovering individuals with useful alleles

Genome scans such as envGWAS are only one plausible way to identify useful genetic diversity in germplasm collections. Rather than using georeferencing data and genomic data to identify specific candidate loci, the same data can be used to identify varieties that likely perform better in distinct environments through genomic prediction and incorporation of environmental predictors. In this study, we compared the effectiveness of using environment-of-origin data to the value of envGWAS-discovered alleles in phenotypic studies.

We present results from our GPoE analyses in which we observe strong genetic differentiation of traditional varieties across environment-of-origin gradients. This work is consistent with gradient forest and other methods in conservation genetics which predict maladapted populations and potential sources of diversity for genome-informed assisted gene flow at broad geographic distances, but struggle at more local scales [22,95]. In the context of breeding, if genotyping every individual in a diversity panel is prohibitively expensive or population structure is unknown, environment-of-origin data of collections may thus enable selection of a starting pre-breeding population. We found that environment-of-origin data of traditional varieties modeled alone explained a small but significant percentage of the variation in yield (Fig 5). This is consistent with extensive previous evidence that maize traditional varieties are adapted along environmental clines [61], as well as previous analysis of a subset of Mexican accessions in this dataset by McLaughlin et al. [96] showing that random forest models utilizing only environment-of-origin data were capable of predicting some root anatomy traits.

However, our study shows that environmental data explained negligible additional variation in field trials beyond what was predictable using genomic data. Population structure, isolation-by-distance, and spatial autocorrelation of environment can generate correlations between environmental values and performance that would be equally captured by genomic data [87,97]. While our GPoE results indicated high correlation between genetics and environment at the continental scale, this association was diminished in our leave-one-population-out spatial sampling. Our results support a body of work that gene-environment association methods applied to collections of natural populations or traditional varieties must

account for population structure vis-à-vis spatial proximity [94,98]. Applications of passport data for selecting environmentally adaptive varieties must take into account the geographic scale of adapted populations and detailed consideration of the environmental clines in question.

In contrast to predictions derived from environmental data or environmentally-associated SNPs, models using whole-genome GBS markers had reasonably high predictive ability for traditional variety crosses. This suggests that given existing field trial data in relevant environments, yield predictions from genome-wide genotyping data likely capture more dimensions of environmental adaptation and selection than would be captured from environmental variables selected *a priori*. However, our results suggest much of the value of genome-wide genotyping may simply be the quantification of population structure. The GBLUP prediction model we used gains accuracy from both historical LD with causal markers and from linkage due to population structure [99,100]. While the first 5 PCs explain less than 5% of genetic variation in our population, we find that the first 5 PCs of the genotyping data provide phenotypic predictive ability similar to the full kinship matrix, suggesting that much of the predictive ability of GBLUP comes from population structure. This paradox that principal component axes explain phenotype more than environment-of-origin may be explained by relatively weak signals of environmental adaptation in our field experiments. Alternatively, high polygenicity in the traits of interest may result in a weaker ability of selected adaptive alleles to explain variation in phenotype. While this has been previously shown to be true in established breeding populations of hybrid temperate maize [101], we show that this occurs in diverse traditional variety collections as well. Morphological variation due to selection for agronomically important traits may be limited to a small percentage of genetic variation that is strongly reflected in the top principal components. Indeed, we believe that this predictive ability is consistent with the top 5 principal components explaining $\sim$ 5% of genetic variation: if predictive accuracy = $r = \sqrt{\text{variance explained}}$, we would estimate 5% of genetic variation explained by PCs would result in a predictive accuracy of $\sim$ 0.22, which appears to be consistent with our results.

In natural populations such as forest management, Capblancq et al [102] suggested that by correcting for population structure to remove false positives, potential true positives heavily confounded with geography and population structure will be removed as well, which may be counterproductive in identifying adaptive material that could be used for recolonization in red spruce. To test this, we removed population structure correction from our envGWAS to increase the number of true discoveries at the cost of increasing the false discovery rate, obtaining 1000 lead SNPs that would tag associations to both climate and population structure and assessed this group in our phenotypic prediction. We observed that uncorrected envGWAS SNPs with PCs only occasionally performed better than PCs alone or against matching random SNPs, and never performed better than the full gBLUP model (S11 Fig).

Thus, learning from which genetically- or geographically-ordained populations a traditional variety or ecotype is collected may provide information to help prioritize which germplasm to introduce into a recolonization or breeding program. However, this is a relatively coarse type of information dependent on the population of interest, and may provide little benefit to breeders or conservationists hoping to leverage the full value of a large germplasm collection.

## Utility of environmental data for pre-breeding and restoration

Unlike Lasky et al. [43], who found that incorporating the top 100-250 representative envGWAS SNPs outperforms kinship-only GBLUP models in panicle weight prediction, our work

did not find a significant change in predicting yield component traits using a limited number of envGWAS SNPs in a GBLUP model.

Our results were in concordance with a previous analysis of this maize dataset for flowering time by Romero-Navarro et al. [50], who found similar predictive ability for flowering time using genome-wide SNPs and a large set of envGWAS markers. In addition, our results were similar to those of Kehel et al. [34] who showed that models including passport environmental data of traditional varieties do not provide a significantly higher predictive ability than models with only genomic data, even with higher SNP density.

Despite not being useful for genetic value predictions in our collections, our results don't preclude envGWAS-identified loci being used in other ways in germplasm collection characterization and pre-breeding. For example, individual envGWAS may still identify functionally relevant alleles that could be incorporated into breeding via marker-assisted selection or genome editing [64]. Additionally, envGWAS loci could be used in breeding by first selecting high-yielding traditional varieties in target environments via genomic prediction, and then ranking the selected varieties based on the enrichment of novel envGWAS loci not already present in elite hybrids. Subsequent mapping populations could then be used to evaluate adaptive envGWAS loci [35], partitioning performance based on parental origin. For conservation or recolonization efforts, environmental data may be combined with genetic and geographic data to predict future locations for introducing new populations, but understanding the underlying architecture and biology of the system is necessary to achieve meaningful gains in fitness [46,95]. Finally, environmental data may continue to be useful for predicting gene-environment interactions where differences in environmental conditions between trial locations can be leveraged to identify gene-by-environment effects and select varieties with favorable responses to specific environmental stresses [43,85].

# Materials and methods

## Samples and genotyping

The Seeds of Discovery project (SeeD) has characterized the CIMMYT International Germplasm Bank maize collection containing over 24,000 accessions of traditional varieties primarily from North, Central, and South America. Here, we used data generated across a core collection of $\approx 4,000$ accessions from this project representing the breadth of environmental variation across the Americas as described in [50]. Accessions were narrowed down to 3511 genotypes representing 2895 accessions based on availability of accurate coordinates for each accession's collection location, where 6-month growing season environment data was complete for all variables of interest (see details in **Environmental data** below).

Briefly, each individual was sequenced using genotyping-by-sequencing [65] to a median 2X coverage. Genotypes were called using TASSEL [103], and imputed using BEAGLE 4 [104] resulting in an initial set of 955,120 imputed SNPs reported in [50] using the B73 v4 reference genome [105].

Genotypes with more than 25% missing data and biallelic SNPs with minor allele frequency > 0.01 were kept after filtering, resulting in a final filtered set of 345,270 SNPs. To estimate kinship, a genomic relationship matrix $K = (X^T X)/(2 \sum p(1-p))$ was estimated using SNPs above MAF > 0.05 for the genotype matrix $X$ as per [106].

PCA analysis was performed using the R package *prcomp* on a subset of 5000 randomly sampled SNPs taken genome-wide from the filtered and imputed GBS data.

## Environmental data

For each accession in our dataset, environment data were obtained using the geographic coordinates attached to collection metadata. Monthly estimates of minimum and maximum temperature and total precipitation were obtained from WorldClim 2 [107], aridity measurements were obtained from the Global Aridity Index and Potential Evapotranspiration Database [108], relative humidity measurements were obtained from ERA5 [109], and elevation for each accession was taken from the SRTM 90m Digital Elevation Database v4.1 [110]. Environmental variables were calculated across 6-month growing seasons for each accession as defined by the growing agroecological zone from the FAO [111].

Variables calculated included: minimum temperature (tmin) over the growing season; maximum temperature (tmax) over the growing season; range of temperature (trange) as defined by tmax – tmin; total precipitation (precipTot) accumulated over the growing season; mean aridity (aridityMean); mean relative humidity (rhMean), and elevation at point of collection for each accession. We include max temperature and minimum temperature as environmental variables as it is known that heat stress affects developmental and physiological traits such as photosynthesis and shortened life cycle [112,113]. Likewise, as precipitation levels such as high rainfall and drought affect maize yields [114], we include total precipitation for our model. As vapor pressure deficit is a major crop model covariate driving yield [115], we include relative humidity and mean aridity measures as an indirect proxy measure. S4 Fig illustrates the distributions and correlations between environmental variables covered by this population. To model non-Gaussian distributions of environmental variables such as precipitation in a non-parametric manner, we calculated the ranked inverse normal transformation (INT) for the whole population using the formula

$$INT(W_i) = \phi^{-1}\left(\frac{\text{rank}(W_i) - c}{N + 1 - 2c}\right)$$

where we transform each environment value $W_i$ using the conventional Blom's offset $c = 3/8$ [116,117]. Transformed environmental data, along with all intermediate results for the analyses below, are available on a Dryad database [118].

## Trials

As part of the SeeD project evaluation of the traditional variety core collection, offspring of traditional variety collections and groups were planted in multiple environments under a replicated F1 crossing design known as F-One Associated Mapping (FOAM) (see [50] for design details). Here, we analyze data on plant height (PH), the total weight of ears (kernels and cob) measured in the field (field weight (FW)), bare cob weight (BCW), and grain weight per hectare adjusted at 12.5% of humidity (GWPH) from [62], as well as flowering traits (days to female flowering (DtF), and anthesis-silking interval (ASI)) from [50].

Two important features of the crossing experiment were included to ensure that phenotype data were not overly biased by elevational adaptation. First, plants were preferentially grown in locations that were of similar adaptation (highland tropical, sub-tropical or lowland tropical) to their home environment. While F1s with a highland accession parent were grown in low elevation and vice versa, on average, more plants from highland accessions were grown in high elevation than low elevation trials. Second, each plant was crossed to a tester that was adapted to the environment that the F1 seeds were grown in, for a total of $\approx 4700$ testcross genotypes (S4 Table shows the number of testers used in each trial). Both of these design features facilitate comparison of a larger sample of accessions, but also lead to an unbalanced

experimental design and reduce apparent adaptive differences among traditional varieties, making our estimates of adaptation more conservative.

Briefly, the F1 offspring of each accession were grown in ambient field conditions across a total of 23 trials in 13 locations in 2 years. Each testcross family was grown in a single plot with a range of 9-26 (average of 16) offspring per family, with an average of 850 accessions (collections and groups from the seed bank) grown in each trial in an augmented row-column design. See experimental design in S1 Fig. As we were only interested in accessions with available geographic coordinate data, we considered only accessions that were labeled as collections: S3 Table shows the number of collection accessions with environmental data measured for each phenotypic trait in each trial, averaging 586 per trial. While this range of field environments (S1 Table) is exceptional with respect to the standard for field based studies attempting to identify the genetic basis of environmental adaptation, we do note that there are a number of maize environments (e.g. extreme high elevation above 2500m, tropical with invariant daylength, and wet, low elevation) that may not be well represented here but should be of interest to future studies of this nature. Across the 23 trials, plants were phenotyped for a variety of agronomically relevant traits (S2 Table). In summary, our field trials included 19,446 plots spanning trial locations, testcrosses, and accessions, with 77,685 phenotypic observations.

## Phenotypic data

Breeding values for each traditional variety were estimated as described in Romero-Navarro *et. al.*[50], controlling for design variables in a complete nested model. Briefly, raw phenotype values were converted into best linear unbiased predictions (BLUPs) that controlled for trial, checks, field position, plot, and tester effects.

To ensure that effect sizes of genetic values have appropriate scale across trials and to correct for shrinkage, BLUPs were deregressed [119]. Trait heritability scores for each trial were calculated as $h^2 = 1 - \frac{\text{mean BLUP variance}}{\text{genetic variance}}$. For each trait analyzed, we kept only trials with heritability scores greater than 0.1 for that trait, and removed trial:trait:tester combinations with less than ten unique values to filter out populations with low sample sizes. In total, 2689 genotypes representing 2518 unique collection accessions with sufficient sample size across trials and had environmental data connected to collection coordinates were used for phenotypic analyses.

## Environmental adaptation

Similarly to Gates et al. [62], we tested for evidence of environmental adaptation by estimating the relationship between the environment-of-origin of each traditional variety and the yield trait values of the testcross families in the field trials. Environmental adaptation is indicated if traditional varieties collected from environments more different from the environment of each trial have lower values of the yield traits [120]. We define environmental adaptation differently from a more strict definition of local adaptation, where potential biotic stresses or other factors specific to a proximal geographic location may influence local vs foreign differences differently than predictable elevational or climatic clines.

To test for this, we fit a statistical model relating the environment-of-origin and the environment of each trial to the variation in yield trait values in each trial. Here, we used elevation, mean temperature, and annual precipitation measurements from WorldClim 2 [107] so that time was consistent between maize collection and trial location coordinates. We fit a quadratic function to trait values within each trial as a function of the environment-of-origin

value of each traditional variety, and modeled the environmental value that maximized this function as a function of the environment of the trial. Specifically, we fit:

$$y_{ijk} = c_{ij} + a_i(x_{ijk} - h_i)^2 + e_{ijk}$$
$$c_{ij} \sim \mathrm{N}(\mu_c, \sigma_a^2), \quad \mu_c \sim \mathrm{N}(0,1)$$
$$a_i \sim \mathrm{N}_-(0,1)$$
$$h_i = \mu_h + X_i\beta_h + \mathrm{N}(0, \sigma_h^2), \quad \mu_h \sim \mathrm{N}(0,1), \quad \beta_h \sim \mathrm{N}(0,3)$$
$$e_{ijk} \sim \mathrm{N}(0, \sigma_{e_i}^2)$$

We fit this model separately for each trait and each candidate environmental variable, where $y_{ijk}$ is a z-scaled de-regressed BLUP of F1 testcross family $k$ crossed to tester $j$ in trial $i$ with climate value of the accession $x_{ijk}$ and climate value of the trial $X_i$. $c_{ij}$ represents the maximum value of each response function, unique for each tester for each trial, $a_i$ is the quadratic parameter specifying the curvature of the response function and constrained to be non-positive through its prior, and $h_i$ is the coordinate of the maximum of the response function in each trial. We modeled $h_i$ as unique for each trial, but with a possible relationship to the climate value of the trial $X_i$, with regression coefficient $\beta_h$. If $\beta_h$ is positive, it implies that trials with higher values on a climate axis select for accessions originating from high climate values. If $\beta_h = 1$ and $\mu_h = 0$, accessions from similar climates to the trial had highest fitness values. Therefore, positive values of $\beta_h$ indicate environmental adaptation, while larger (negative) values of $a_i$ indicate stronger decreases of fitness as accessions' environment-of-origin deviates further from the optimum $\beta_h$. We implemented this model using the R package *brms* to obtain posterior values for these parameters [121].

We find that in most trial and environmental variable combinations, environmental adaptation appears relatively weak but reflects a negative curvature in the quadratic parameter because varieties from extreme values show reduced fitness. Models permitting non-negative values for $a_i$ were assessed in S1 File, where we find that a majority of trials in various environments are consistent with negative curvature. In some cases where trials appear to show transfer-distance functions with positive curvature, this can sometimes be explained by an insufficient range of environment-of-origin values to observe the extreme values (e.g. negative values for precipitation or elevation above 4000 masl). Therefore, we believe that constraining curvatures to be negative is reasonable for these data and do not likely strongly impact our conclusions.

## Genomic Prediction of Environment (GPoE)

GPoE analyses to predict environmental variables from genotypic data ($E \sim G$) was performed with MegaLMM [67]. As no phenotypic data was necessary for this analysis, all 3511 genotypes that had both available genomic data and associated environmental data were used.

Briefly, MegaLMM decomposes the covariance of genetic random effects and environmental variables simultaneously to de-correlate environmental relationships and genetic relationships, and uses those covariance matrices to estimate withheld testing data. The kinship matrix $K$ was calculated as the genomic relationship matrix above and served as the input for genotype data. The model for prediction was $\mathbf{Y} = \mathbf{F}\Lambda + \mathbf{U}_R + \mathbf{E}_R$ where $\mathbf{Y}$ represents a matrix of environmental values for 3511 genotypes and seven INT-transformed environmental variables, $\mathbf{F}$ and $\Lambda$ are factor scores and loadings for $k = 7$ latent factors, and $\mathbf{U}_R$ and $\mathbf{E}_R$ are matrices of residual genetic and non-genetic values not explained by the factors. $\mathbf{F}$ was also modeled as a function of genetic and non-genetic effects: $\mathbf{F} = \mathbf{U}_F + \mathbf{E}_F$. Matrices $\mathbf{U}_R$ and $\mathbf{U}_F$ were

matrix-normal random variables with diagonal column covariances and row covariances proportional to $\mathbf{K}$. Matrices $\mathbf{E}_R$ and $\mathbf{E}_F$ were matrix-normal random variables with diagonal row and column covariances. MegaLMM was run with default parameters [67] with a burn-in of 2000 iterations, and convergence reached over another 2000 iterations.

K-fold cross-validation was performed for $k = 10$ folds, with a testing set of approximately 350 accessions having environmental data withheld before training. Training was done on non-withheld data through MCMC sampling via MegaLMM: 200 samples were collected each iteration. After 20 iterations of sampling, predicted $\hat{\mathbf{U}}$ values for each environmental variable was calculated as the posterior mean for both testing and training set accessions: $\hat{\mathbf{U}} = \mathbf{U_R} + \mathbf{U_F}\Lambda$ To evaluate predictive ability, Pearson's $r^2$ correlation for each environmental variable was calculated as $\mathbf{cor}(\hat{\mathbf{U}}, \mathbf{Y})$

To test the effect of geographic distance on genotype-environment correlations, we used the R package *spatialsample* [122] to split our study population into ten geographically separated folds, clustered non-deterministically based on distance. Cross-validation was then performed as described above for our $k = 10$ spatial folds, withholding accessions for a given fold with a buffer distance of 400 km to prevent accessions directly bordering the fold to inform prediction for the accessions within the fold, thereby enforcing spatial independence. A pairwise t-test was performed to test whether a significant difference between random and spatial k-fold cross-validation existed for each environmental variable and trial combination.

## Environmental GWAS (envGWAS)

Environmental genome-wide associations (envGWAS) were run for each environmental variable (elevation, tmax, tmin, trange, rhMean, precipTot, aridityMean). The same 3511 genotypes that had both genomic data and associated environmental data as in the GPoE analysis were used for envGWAS. Associations were made between the filtered SNP set and INT-transformed growing season data described above.

envGWAS was performed using the R package *JointGWAS* [68]. JointGWAS tests each SNP for association with variation in *any* environmental variable using an F-test. Residual correlations among environmental variables and correlated genetic background effects across genotypes were accounted for using genetic and residual covariance matrices estimated using MegaLMM [67]. Markov-Chain Monte Carlo sampling to obtain variance estimates for each variable was performed with a burn-in of 2000 iterations, and convergence reached over another 2000 iterations.

The association model was run in JointGWAS for each SNP as follows:

$$\mathbf{Y} = \mathbf{XB} + \mathbf{E}$$

where for $\boldsymbol{n}$ genotypes, $\boldsymbol{t}$ environmental variables, and $\boldsymbol{b}$ number of genotyped SNPs, $\boldsymbol{Y}$ was the $\boldsymbol{n} \times \boldsymbol{t}$ matrix of environmental values, $\boldsymbol{X}$ was the $\boldsymbol{n} \times \boldsymbol{b}$ genotype design matrix, and $\boldsymbol{B}$ was the $\boldsymbol{b} \times \boldsymbol{t}$ matrix of estimated genotype effects.

$\mathbf{E}$ was modeled as a random variable with multivariate normal distribution with mean zero and covariance $\mathbf{Z}(\mathbf{G} \otimes \mathbf{K} + \mathbf{R} \otimes \mathbf{I})\mathbf{Z}^T$, where $\mathbf{G}$ and $\mathbf{R}$ are background genetic and residual covariance matrices among environmental variables estimated as the posterior means of these matrices from MegaLMM run with default parameters, $\mathbf{K}$ is the genomic relationship matrix computed above, $\mathbf{I}$ an identity matrix, and $\mathbf{Z}$ represented the design matrix relating all accession:environmental value combinations to the observed environmental values. For computational speed, JointGWAS pre-multiplies both sides of the association model by the inverse of the Cholesky combination of the residual covariance matrix, turning the model into

a standard linear model, after which a normal F-test was applied to the coefficient of the **X** term.

To account for population structure and inter-chromosomal LD, the genomic relationship matrix **K** was separately calculated for each chromosome and the association test run separately per chromosome as well. A p-value significance threshold of $10^{-5}$ was used to identify SNPs that were considered significant to any environmental variable, set at a slightly lower threshold than the Bonferroni correction to evaluate more loci for phenotypic enrichment and prediction i.e. for pre-breeding introgression.

To evaluate SNP association without population structure correction, we ran the above analysis using the identity matrix in lieu of the kinship matrix $K$. We then identified SNPs based on a p-value threshold of $10^{-250}$, clumped them to obtain a set of 1000 lead SNPs and a set of random SNPs of matching allele frequency, and evaluated them in phenotypic prediction (see methods below).

## Phenotypic association

In addition, phenotypic genome-wide associations were run for each trait (PH, FW, BCW, GWPH, DtF, ASI) across all trials using JointGWAS. Here, we test whether each SNP has association with phenotypic variation in *any* trial with an F-test.

As in the envGWAS, genetic and residual covariance matrices were estimated using MegaLMM, but accounting for residual correlations among trials and genetic background. Since we measured the phenotypes between F1 offspring of our accessions, we divide our genomic relationship matrix $K$ by a factor of 4 to account for half-sib kinship. The same default parameters for MegaLMM were used as in the envGWAS.

The association model for each SNP was:

$$\mathbf{Y} = \text{Trial:Tester} + \text{Trial:}\mathbf{X} + \mathbf{X} + \mathbf{E}$$

where **Y** was the matrix of BLUP values over all trials, Trial:Tester were fixed effects, **X** was the main effect of a SNP, Trial:**X** was the interaction between each SNP and each trial. **E** was modeled the same as in the envGWAS above, but where **G** and **R** are background genetic and residual covariance matrices among trials from MegaLMM, and **Z** a design matrix relating all possible accession:trial combinations (columns) to the observed combinations (rows). To increase GWAS power, when testing each SNP we excluded all SNPs on the same chromosome when calculating **K** for the covariance computation. The genome-wide **K** was used with MegaLMM when estimating **G** and **R**.

Finally, normal F-tests were applied to the coefficients of both the terms **X** and Trial:**X**. The p-values from these two F tests were combined using Fisher's method into a single test of any association of the marker in any of the trials. This was done as previous analyses separating the global $G$ and $G \times E$ terms found little evidence of identifying interaction effects, and so we chose to combine the terms together to increase discovery power.

## Enrichment of phenotypic consequence for envGWAS SNPs

From our envGWAS, we take the top SNPs most significant for environment and observe if those SNPs similarly had significantly higher effect on yield or developmental traits from our phenotypic trials. To prevent over-representation from envGWAS peaks with large genomic regions encompassing many adaptive loci such as *Inv4m*, we opted to measure the effect of a given representative SNP obtained through LD clumping. Clumping was performed genome-wide to capture trans-LD across chromosomes as well as to identify mis-mapping in the

assembly. LD clumping was performed by calculating the pairwise $r^2$ of all SNPs above the $10^{-5}$ p-value significance threshold. We wrote a custom Python script to perform clumping using a greedy algorithm, prioritizing the most significant SNPs as lead SNPs and assigning SNPs to a lead SNP if they had correlation $r^2 > 0.3$ to a lead SNP. Under these parameters, we obtained 32 lead SNPs representing top peaks in our envGWAS results.

Enrichment analysis of our envGWAS SNPs in our phenotypic GWAS was performed by taking our top 32 SNPs and obtaining 1000 samples of 32 random SNPs across the genome with minor allele frequency (MAF) matching the distribution of that of our envGWAS SNPs. For each phenotypic trait, we pulled the $-log$(p-value) from the phenotypic GWAS for both our lead envGWAS SNPs and each set of matching MAF SNPs. We then compared whether the average $-log$(p-value) for the envGWAS SNPs was an outlier compared to the null distribution of randomly sampled SNPs. P-values were calculated by finding the fraction of matching SNP sets with a larger average $-log$(p-value) compared to the envGWAS SNP set.

## Candidate genes

To identify candidate genes from our envGWAS, we use the 32 lead SNPs obtained above from LD clumping, such that SNPs significant for environment are grouped together with SNPs within LD. Candidate genes were then chosen based on nearest genomic distance to lead clumped SNPs using BEDtools [123] against the gene model annotation of the V4 assembly of the maize genome [105]. We then filtered down genes to only that had curated annotations and known models in the V4 assembly of the B73 reference genome. LD analysis was performed by comparing $r^2$ of all SNPs in the GBS sequencing within a 300kbp upstream and downstream range of a given lead SNP, with the exception of S4_177835031, in which we used the 600kbp range.

## Genomic prediction (GP)

We performed genomic prediction on the deregressed phenotypic BLUP data, using the accessions with available climate data. We predicted the genomic BLUP (gBLUP) value for each accession in our panel and compared it to the observed BLUP value from field trials for each phenotypic trait.

As each testcross' performance is partially controlled by the tester's genotype, which was non-randomly assigned with respect to both traditional variety genotype and environment-of-origin, we estimated the predictive ability of each model by calculating the Pearson's correlation between predicted and observed values within sets of testcrosses made with the same tester. We did this using cross-validation by assigning all testcrosses made with the same tester to the same fold, and trained models on yield BLUP residuals (corrected for tester ID) from all other testcrosses. Models were fit and evaluated separately for each trial.

Linear prediction models were constructed in rrBLUP [124] with the general model $Y = X\beta + Zu + \epsilon$ where $Y$ represented the BLUP residuals of each sampled individual, correcting for tester ID. For all models, a design matrix $X$ of tester genotypes remaining in the training set was included as a fixed effect. Principal components were calculated using the singular value decomposition of the kinship matrix for the accessions present in each trial. The first five eigenvectors were included as features in each model.

We constructed the following models:

1. A baseline PCs-only model where $X$ included tester and the first five principal components.

2. A PCs + envGWAS SNPs model where we added a $3511 \times 32$ random effect matrix $Z$ of the lead SNPs representing the 32 loci significantly associated with environment from our envGWAS analysis.

3. A PCs + random SNPs model where we additionally created an equivalent $3511 \times 32$ random effect matrix $Z$ matrix of randomly sampled SNPs from the genome with matching allele frequency to the envGWAS SNPs.

4. An all-SNPs model where the random effect $u$ had covariance $K$, the kinship matrix calculated from all SNP markers in the GBS data, in addition to the $X$ fixed effects matrix of tester and first five PCs.

5. An all-SNPs + environment model that was identical to the previous, but also included the INT-transformed environment-of-origin data for each accession as fixed effects in the $X$ design matrix.

Further, to assess how how well phenotypic variation could be predicted by environmental data, we additionally tested models given only aggregate environment data, and with models employing non-parametric random forest models using the R package RandomForest [125] with ntree = 1000. Random forests were run with 1000 decision trees. These included:

6. A linear model in rrBLUP with a $X$ matrix with only tester + climatic data;

7. A random forest with tester + climatic data as predictor variables;

8. A random forest with tester + climatic data + first five PCs as predictor variables.

To measure differences in predictive ability between the first five models, we fit the following linear mixed model to the correlation measures of each model in each fold using the *R* package *lme4* [126]:

$$r \sim \text{trial} + \text{model} + (1—\text{trial:model}) + (1—\text{trial:fold}) + e$$

where the formalism (1—X) represents a random effect with grouping level specified by the factor X. Since the sampling variance of Pearson's r is higher for smaller sample sizes, residuals $e$ were weighted by the inverse of the sample size. Comparisons among the main effects of the models were evaluated using Tukey's method implemented in the *emmeans* and *lmerTest* packages [127,128].

## Supporting information

**S1 Table. Trial metadata.** Location, year, coordinates, and elevation of each location held for phenotypic field trials.
(CSV)

**S2 Table. Accessions measured for each trait per trial.** Number of accessions in each trial (location and year) measured for each trait.
(CSV)

**S3 Table. Testcrosses per trial.** Number of accessions in each trial crossed to each tester, before filtering.
(CSV)

**S4 Table. Testers per trial.** Number of unique testers used for crossing and measured for traits in each trial.
(CSV)

**S5 Table. Number of trials where testers were used.** Number of trials where given elite testers were used for crossing and measured for each trait, before filtering.
(CSV)

**S6 Table. Correlation matrix between PCs and environmental variables.** Correlation between top 5 PCs from SNP data and growing season climate data (tmin, tmax, trange, precipTot, aridityMean, rhMean, elevation) as well as latitude.
(CSV)

**S1 File. All transfer plot model curves for each trial, unconstrained curvature.** All transfer plot model curves, where model is unconstrained to be negative. Panels represent fitness curves for individual trials, where individual points represent scaled BLUPs of an accession grown in that trial, plotted against an environmental variable (mean temperature, annual precipitation, elevation of origin). Each page contains panels for all trials where a phenotypic trait was measured, against a single environmental variable.
(PDF)

**S1 Fig. Experimental design.** Flowchart visualizing experiment design, including filtering of initial germplasm from CIMMYT, number of testcrosses, plots in locations, and number of observations and genotypes.
(TIF)

**S2 Fig. Biplot between genotypic PCA1 and PCA2.** Biplot of PCA1 (representing 3.76% of genetic diversity) vs PCA2 (representing 0.92% of genetic diversity) for all accessions' genotypes. Points are colored by latitude of collection.
(TIF)

**S3 Fig. Scree plot of genotypic PCA.** Scree plot of relative contribution of top 10 PCs towards genetic variation in this population.
(TIF)

**S4 Fig. Distributions and correlations between environmental variables.** Distributions and correlations between non-INT-transformed growing season environment variables for accessions used in GEA analysis.
(TIF)

**S5 Fig. All transfer plot model curves, aggregated over trait and environmental variable.** All transfer plot model curves for selected yield component phenotypic trait and environmental variable combinations. Panels **a–c** measure BLUP residuals for bare cob weight, **d–f** for field weight, **g–i** for corrected grain weight, and **j–l** for plant height.
(TIF)

**S6 Fig. All regression plots of transfer plot optimal values.** All regression plots of optimal environmental value for a given trial against observed environmental value. Panels **a, d, g j** measure elevation, **b, e, h k** measure precipitation, **c, f, i, l** measure temperature. Individual dots represent trials with measurements for the specific combination.
(TIF)

**S7 Fig. Map of spatial GPoE folds and predictive ability by fold.** Correlations between genetics and environment are demonstrated through prediction but decrease when considering spatial distance. **A)** Map of sampled cross-validation folds used in spatial GPoE. (**B–E)** Pearson's *r* correlation and RMSE for spatial (B, D) and random (C, E) sampling for each environmental variable across folds.
(TIF)

**S8 Fig. Predictive ability for all phenotypic traits, separated by trial.** Predictive ability results for field weight, corrected grain weight per hectare, bare cob weight, and plant height, where panels are separated by trial (location and year), measured as Pearson's r correlation to observed BLUP value. Predictions were made separately for each set of accessions crossed to a given tester for each trial (individual points). Models tested here included top five PCs, random SNPs + PCs, envGWAS SNPs + PCs, all SNPs, and all SNPs + climate data in a linear model.
(TIF)

**S9 Fig. Predictive ability for all phenotypic traits, environmental variable predictive models.** Environmental data model predictive ability results for field weight, corrected grain weight per hectare, bare cob weight, and plant height, where panels are separated by trial (location and year). Models tested here included top five PCs, climate data via linear models, climate data modeled with random forests, and climate data + top five PCs with random forests.
(TIF)

**S10 Fig. QQplot of envGWAS.** Quantile-quantile plot of multivariate joint envGWAS to assess inflation of p-values. Observed p-value on y-axis plotted against expected p-value on x-axis, with one-to-one line in red.
(TIF)

**S11 Fig. Results of envGWAS without population structure correction. (A)** Results from envGWAS without incorporating kinship matrix to control for population structure. **(B)** Results comparing phenotypic prediction from using top 1000 lead SNPs from unstructured envGWAS + PCs to models including only top 5 PCs (population structure), all SNPs (gBLUP), and a model including random 1000 SNPs of matching allele frequency + PCs.
(TIF)

**S12 Fig. Spatial distributions of candidate SNPs.** Maps of lead SNP alleles representing **(A)** the *Inv4m* inversion and the **(B)** *hsftf9* putative candidate locus. Colors represent number of alternate allele present for given collection accession. Base layer map tiles are from Stamen Design under CC BY 4.0, accessed via the R package ggmap [66]. Data by OpenStreetMap, under ODbL.
(TIF)

**S13 Fig. Linkage disequilibrium plots of SNPs near lead SNPs.** Comparison of LD for all SNPs within ±300kb of top 32 lead SNPs. SNPs found significant in envGWAS are colored in red. For S4_177835031, 600kb was used to describe *Inv4m*
(TIF)

## Acknowledgments

This research collaboration was made possible through the CIMMYT Seeds of Discovery project. CIMMYT would like to acknowledge the Sustainable Modernization of Traditional Agriculture (MasAgro) project supported by the Ministry of Agriculture and Rural Development (SADER) of the Government of Mexico for funding the Seeds of Discovery maize traditional variety characterization.

## Author contributions

**Conceptualization:** Dan J. Gates, Edward S. Buckler, Matthew B. Hufford, Garrett M. Janzen, Rubén Rellán-Álvarez, Fausto Rodríguez-Zapata, Ruairidh J. H. Sawers, Sarah J. Hearne, Jeffrey Ross-Ibarra, Daniel E. Runcie.

**Data curation:** Forrest Li, Dan J. Gates, J. Alberto Romero Navarro, Kai Sonder, Martha C Willcox, Daniel E. Runcie.

**Formal analysis:** Forrest Li, Daniel E. Runcie.

**Funding acquisition:** Edward S. Buckler, Sarah J. Hearne, Jeffrey Ross-Ibarra, Daniel E. Runcie.

**Investigation:** Forrest Li, Dan J. Gates, Garrett M. Janzen, Rubén Rellán-Álvarez, Fausto Rodríguez-Zapata, Ruairidh J. H. Sawers, Samantha J Snodgrass, Martha C Willcox, Daniel E. Runcie.

**Methodology:** Forrest Li, Jeffrey Ross-Ibarra, Daniel E. Runcie.

**Project administration:** Sarah J. Hearne, Jeffrey Ross-Ibarra.

**Resources:** Sarah J. Hearne, Jeffrey Ross-Ibarra.

**Software:** Jeffrey Ross-Ibarra.

**Supervision:** Sarah J. Hearne, Jeffrey Ross-Ibarra, Daniel E. Runcie.

**Visualization:** Forrest Li, Daniel E. Runcie.

**Writing – original draft:** Forrest Li, Dan J. Gates, Jeffrey Ross-Ibarra, Daniel E. Runcie.

**Writing – review & editing:** Forrest Li, Edward S. Buckler, Matthew B. Hufford, Garrett M. Janzen, Rubén Rellán-Álvarez, Fausto Rodríguez-Zapata, Ruairidh J. H. Sawers, Samantha J Snodgrass, Sarah J. Hearne, Jeffrey Ross-Ibarra, Daniel E. Runcie.

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
