## [Decision Letter · Decision Letter 0]

PGENETICS-D-24-01188

The utility of environmental data from traditional varieties for climate-adaptive maize breeding

PLOS Genetics

Dear Dr. Li,

Thank you for submitting your manuscript to PLOS Genetics. After careful consideration, we feel that it has merit but does not fully meet PLOS Genetics's publication criteria as it currently stands. Therefore, we invite you to submit a revised version of the manuscript that addresses the points raised during the review process.

Please submit your revised manuscript within 60 days Feb 22 2025 11:59PM. If you will need more time than this to complete your revisions, please reply to this message or contact the journal office at plosgenetics@plos.org. Please include the following items when submitting your revised manuscript:

We look forward to receiving your revised manuscript.

Kind regards,

Angela Hancock, Ph.D.

Academic Editor

PLOS Genetics

Justin Fay

Section Editor

PLOS Genetics

Aimée Dudley

Editor-in-Chief

PLOS Genetics

Anne Goriely

Editor-in-Chief

PLOS Genetics

**Additional Editor Comments :**

Thank you for submitting your manuscript to PLoS Genetics! It has now been assessed by two expert reviewers. Both are very positive and believe the study will be a useful addition to the literature, but they also mentioned some concerns and provided suggestions for improvement. Many of the reviewers' comments are aimed at making the manuscript more accessible to a broader range of readers, but they also asked for a couple of additional analyses to improve robustness of the results. In your revision, please address all concerns from the reviewers.

**Journal Requirements:**

At this stage, the following Authors/Authors require contributions: Forrest Li, Dan J Gates, Edward S Buckler, Matthew B Hufford, Garrett M Janzen, Rubén Rellán-Álvarez, Fausto Rodríguez-Zapata, J. Alberto Romero Navarro, Ruairidh JH Sawers, Samantha J Snodgrass, Kai Sonder, Martha C Willcox, Sarah J Hearne, Jeffrey Ross-Ibarra, and Daniel E Runcie. Please ensure that the full contributions of each author are acknowledged in the "Add/Edit/Remove Authors" section of our submission form.

The list of CRediT author contributions may be found here: https://journals.plos.org/plosgenetics/s/authorship#loc-author-contributions

Potential Copyright Issues:

i) Figures 1, and S10. Please (a) provide a direct link to the base layer of the map (i.e., the country or region border shape) and ensure this is also included in the figure legend; and (b) provide a link to the terms of use / license information for the base layer image or shapefile. We cannot publish proprietary or copyrighted maps (e.g. Google Maps, Mapquest) and the terms of use for your map base layer must be compatible with our CC BY 4.0 license.

7) Please ensure that the funders and grant numbers match between the Financial Disclosure field and the Funding Information tab in your submission form. Note that the funders must be provided in the same order in both places as well. Currently, the order of this funder" Agricultural Marketing Service" is different in both places.

Please indicate by return email the full and correct funding information for your study and confirm the order in which funding contributions should appear. Please be sure to indicate whether the funders played any role in the study design, data collection and analysis, decision to publish, or preparation of the manuscript.

8) Thank you for providing us with your Data Availability statement. We noted that this link "github.com/liforrest/SeeD" reaches a 404 error page. Please amend this to a new link or provide further details to locate the data.

**Reviewers' comments:**

Reviewer's Responses to Questions

Reviewer #1: This study evaluated how well genomic and environmental data can be used to predict maize traits, with the goal of evaluating frameworks for identifying traditional varieties into maize breeding programs. The authors are to be commended for pulling together this extensive dataset and for the most part the manuscript was well written. I hope the authors find the following recommends helpful to improve the presentation and rigor of the study.

- The question of how well the environment-of-origin predicts success of a genotype in the field has potential to be very interesting not only to crop researchers, but also to other selective breeding programs (e.g., bovine, aquaculture) and to restoration. The introduction of the manuscript, however, is focused entirely on crops, and the discussion is focused almost entirely on maize. This focus makes the manuscript more suitable for a specialized journal. Majorly revising the intro and discussion to highlight the importance of this topic across multiple different applications would make the manuscript more suitable for a broader PLOS genetics audience.

- The main conclusion was that despite a clear signal of local adaptation, neither environment-of-origin nor specific climate-associated SNPs predicted trait yields as well as the top 5 PCs. This seems like a paradox that is not fully addressed in the Discussion. The paradox could potentially be explained by either (1) local adaptation not be as strong as suggested, or (2) these traits being highly polygenic, and thus making it hard to uncover the genetic basis of climate adaptation (e.g. https://www.pnas.org/doi/10.1073/pnas.2220313120). It is also interesting that PCs have this much predictive power despite explaining less than 5% of the genetic variation, and this point should be directly addressed in the discussion.

- How strong is local adaptation in maize? This is hard to glean from the manuscript for a few reasons. First, the experimental design is intractable from the writing. From the methods, the design sounds highly imbalanced (a fact that the authors were straighforward about), and the use of vague terms such as “trials”, “test cross genotypes” “tester”, “locations”, “plots” and “accessions” make it hard to follow the design. Defining these terms for readers would be helpful. Also, a conceptual figure that encapsulates the experimental design and methods would greatly benefit the manuscript and help it be accessible to a wider audience. While the crossing design might have reduced the signal of adaptation (making estimates of adaptation more conservative), the prior used on the quadratic functions could have made adaptation look stronger than it really is. The curvature of the quadratic function was constrained to be negative through its prior, which is why all the curves on Figure 2 look flat or negative. I would be hesitant to accept this manuscript (1) without a less constrained analysis on the curvature prior and (2) without a figure 2 that shows both outcome variability and statistical uncertainty (currently only the latter is visualized, https://www.pnas.org/doi/10.1073/pnas.2302491120). I suspect that such a figure would be messy, but that would be the point - there is local adaptation, but there is a lot of variability, which is why environment-of-origin does not offer much predictive power.

- Some minor edits would greatly improve the interpretability of the results. For instance, while it’s straighforward to think about the optimum environment for a given accession/genotype, it’s confusing to think about an “optimum environment for a trial”. It’s also confusing to interpret the “slope of the regression of the apex” and why positive B_h indicates local adaptation - it seems that larger negative values of a_i would be more interpretable. I also found the “spatial CV GPoE” model hard to understand until I read the methods - maybe you mean “leave-one-population-out”? The CV results would also be interesting to a broader audience, as they suggest that predictive values decline rapidly with extrapolation to new populations/environments, and I recommend adding some more on this to the discussion.

Overall, this is an extensive analysis that challenges some of the conventional wisdom regarding the predictive power of environment-of-origin. With revisions that make the manuscript accessible to a wider audience, this will make a nice contribution to the literature.

Reviewer #2: The authors have conducted an impressively scaled study of local adaptation in maize and the phenotypic and environmental associations present in the genome. The methods are well considered and generally rigourous, the results are interesting, and the manuscript is clearly written. It is well worth exploring whether the environmental GWAS approach can outperform a "naive" genomic selection approach when applied to a breeding context. I think this is a very strong candidate for publication in PLOS Genetics, but there are a few areas I'd be interested to see incorporated or more clearly described, as discussed below. I think much more interesting biology could be studied in terms of the relevance to understanding evolution, but perhaps that is planned for a future paper -- I can see that might fall outside of the main focus of the present work. I would recommend minor revisions, but would particularly be interested in seeing major comment #2 being included in the final paper.

Major comments:

1. Line 620-622: while I can see that this step combining terms X and Trial:X is simpler, it perhaps obscures very different patterns and causalities -- SNPs that are involved in local adaptation would be expected to show a significant interaction term for a trait like height, whereas SNPs that were universally beneficial would show a significant main effect. Did you explore whether any interesting signal could be ascertained by studying these separately rather than combining them? Also, when associations with phenotypic traits are discussed on line 225-228, is this for the same Fisher-combined probability of X and Trial:X? Also, I'm assuming that the analysis of the 32 loci in terms of genomic prediction is accounting for the fact that these loci may have had significant main or interaction terms for how well they are able to predict phenotype? I couldn't tell from the methods on lines 624-641 but I think based on it just reimplementing the phenotypic GWAS (line 637) then it is accounting for this (?).

2. It would be interesting to know how well environmental GWAS method would have worked for the genomic selection methods if the authors had not performed population structure correction. For the reasons the authors outline on line 360, using population structure correction can reduce false positives but at the expense of incurring false negatives (i.e. missing some true positives that covary strongly with population structure). But if you are comparing a set of associated SNPs to all the SNPs in the genome for the purposes of genomic selection (line 255), it hardly matters if the set of associated SNPs includes some false positives that are not actually causal but just associated. I suspect this wouldn't perform any better, but it would be a more convincing assessment of whether the methodologically more involved environmental GWAS approach can do better than naive genomic selection with lots of SNPs.

Minor comments:

- Figure 2E: seems a bit strange to show negative precipitation on the y-axis.

- line 587: what are t, b, and n? Please define.

- Line 595: I have no idea what the Cholesky combination does, I can't comment on that technical detail, or how it behaves.

- Line 318: while large effect alleles may be kept rare under purifying/stabilizing selection, they are expected to be enriched under sustained migration-selection balance. See review by Yeaman (2022; Genetics). Similarly, local adaptation is expected to greatly extend blocks of LD around selected loci due to persistent ongoing hitchhiking that never reaches completion (line 320).

**Have all data underlying the figures and results presented in the manuscript been provided?**

Reviewer #1: Yes

Reviewer #2: Yes

PLOS authors have the option to publish the peer review history of their article (what does this mean?). If published, this will include your full peer review and any attached files.

Reviewer #1: No

Reviewer #2: No

**Figure resubmission:**
---

## [Decision Letter · Decision Letter 1]

PGENETICS-D-24-01188R1

Environmental data provide marginal benefit for predicting climate adaptation

PLOS Genetics

Dear Dr. Li,

Thank you for submitting your manuscript to PLOS Genetics. After careful consideration, we feel that it has merit but does not fully meet PLOS Genetics's publication criteria as it currently stands. Therefore, we invite you to submit a revised version of the manuscript that addresses the points raised during the review process.

Please submit your revised manuscript within 30 days May 25 2025 11:59PM. If you will need more time than this to complete your revisions, please reply to this message or contact the journal office at plosgenetics@plos.org. Please include the following items when submitting your revised manuscript:

We look forward to receiving your revised manuscript.

Kind regards,

Angela Hancock, Ph.D.

Section Editor

PLOS Genetics

Justin Fay

Section Editor

PLOS Genetics

Aimée Dudley

Editor-in-Chief

PLOS Genetics

Anne Goriely

Editor-in-Chief

PLOS Genetics

**Additional Editor Comments:**

The reviewers agreed the revised version is now in good shape, but there are a couple of minor outstanding issues regarding clarity of text in one section and figure as well as data availability. These are relatively minor, but the paper could be helped by trying to address them before we make a final decision on the paper.

**Reviewers' comments:**

Reviewer's Responses to Questions

**Comments to the Authors:**

Reviewer #1: The authors have done a satisfactory job addressing the reviewer comments. The dataset and analyses are impressive, and this paper will inform important debates about the utility of envrGWAS SNPs in management.

As a minor comment, the models and results presented on lines 140-155 are still confusing (as well as the relevance of “positive slope parameter” and BH = 1).

It would be helpful in Figure 2A and B to show the trial elevation in addition to the optimum elevation.

Reviewer #2: The authors have responded appropriately to all my suggestions, and I appreciate the addition of the new analysis. I have no further concerns about the manuscript and think this is a good candidate for publication.

I checked the data availability statement and the link for the github repo was broken, and the github author's page did not appear to have a corresponding repo. I think that open data is critical for science, and would suggest the authors publish an archive on dryad or similar repository including all of the intermediate data used to generate each figure, along with documentation and metadata required to interpret them. This is a bit of work, but is an important step, beyond the raw sequence data/climate data.

**Have all data underlying the figures and results presented in the manuscript been provided?**

Reviewer #1: None

Reviewer #2: **No: **See comments to authors

PLOS authors have the option to publish the peer review history of their article (what does this mean?). If published, this will include your full peer review and any attached files.

Reviewer #1: No

Reviewer #2: No

**Figure resubmission:**
---

## [Editor Report · Decision Letter 2]

Dear Dr Li,

We are pleased to inform you that your manuscript entitled "Environmental data provide marginal benefit for predicting climate adaptation" has been editorially accepted for publication in PLOS Genetics. Congratulations!

Yours sincerely,

Angela Hancock, Ph.D.

Section Editor

PLOS Genetics

Justin Fay

Section Editor

PLOS Genetics

Aimée Dudley

Editor-in-Chief

PLOS Genetics

Anne Goriely

Editor-in-Chief

PLOS Genetics

Comments from the reviewers (if applicable):

**Data Deposition**

http://datadryad.org/submit?journalID=pgenetics&manu=PGENETICS-D-24-01188R2

**Press Queries**

---

## [Editor Report · Acceptance letter]

PGENETICS-D-24-01188R2

Environmental data provide marginal benefit for predicting climate adaptation

Dear Dr Li,

We are pleased to inform you that your manuscript entitled "Environmental data provide marginal benefit for predicting climate adaptation" has been formally accepted for publication in PLOS Genetics! Your manuscript is now with our production department and you will be notified of the publication date in due course.

With kind regards,

Anita Estes

PLOS Genetics

On behalf of:
